# Segmentation-Based Measurement of Orbital Structures: Achievements in Eyeball Volume Estimation and Barriers in Optic Nerve Analysis

**DOI:** 10.3390/diagnostics14232643

**Published:** 2024-11-23

**Authors:** Yong Oh Lee, Hana Kim, Yeong Woong Chung, Won-Kyung Cho, Jungyul Park, Ji-Sun Paik

**Affiliations:** 1Department of Industrial and Data Engineering, Hongik University, Seoul 04066, Republic of Korea; yongoh.lee@hongik.ac.kr; 2Department of Computer Engineering, Hongik University, Seoul 04066, Republic of Korea; 3Department of Ophthalmology and Visual Science, The Catholic University of Korea, St. Vincent’s Hospital, Suwon 16247, Republic of Korea; 4Department of Ophthalmology and Visual Science, The Catholic University of Korea, Uijeongbu St. Mary’s Hospital, Uijeongbu 11765, Republic of Korea; 5Department of Ophthalmology and Visual Science, The Catholic University of Korea, Seoul St. Mary’s Hospital, Seoul 06591, Republic of Korea; 6Department of Ophthalmology and Visual Science, The Catholic University of Korea, Yeouido St. Mary’s Hospital, Seoul 07345, Republic of Korea

**Keywords:** deep learning-based segmentation, orbital CT, automated measurement, eyeball volume

## Abstract

Background/Objective: Orbital diseases often require precise measurements of eyeball volume, optic nerve sheath diameter (ONSD), and apex-to-eyeball distance (AED) for accurate diagnosis and treatment planning. This study aims to automate and optimize these measurements using advanced deep learning segmentation techniques on orbital Computed Tomography (CT) scans. Methods: Orbital CT datasets from individuals of various age groups and genders were used, with annotated masks for the eyeball and optic nerve. A 2D attention U-Net architecture was employed for segmentation, enhanced with slice-level information embeddings to improve contextual understanding. After segmentation, the relevant metrics were calculated from the segmented structures and evaluated for clinical applicability. Results: The segmentation model demonstrated varying performance across orbital structures, achieving a Dice score of 0.8466 for the eyeball and 0.6387 for the optic nerve. Consequently, eyeball-related metrics, such as eyeball volume, exhibited high accuracy, with a root mean square error (RMSE) of 1.28–1.90 cm^3^ and a mean absolute percentage error (MAPE) of 12–21% across different genders and age groups. In contrast, the lower accuracy of optic nerve segmentation led to less reliable measurements of optic nerve sheath diameter (ONSD) and apex-to-eyeball distance (AED). Additionally, the study analyzed the automatically calculated measurements from various perspectives, revealing key insights and areas for improvement. Conclusions: Despite these challenges, the study highlights the potential of deep learning-based segmentation to automate the assessment of ocular structures, particularly in measuring eyeball volume, while leaving room for further improvement in optic nerve analysis.

## 1. Introduction

Recent advancements in artificial intelligence (AI) for medical image analysis have significantly improved diagnostic and therapeutic practices, with techniques such as classification, object detection, and segmentation [1,2]. For instance, deep learning models have been employed to classify breast cancer from mammograms with high accuracy, aiding early diagnosis [3]. Object detection algorithms have been used to identify and localize tumors in Computed Tomography (CT) scans, enhancing surgical planning and precision [4]. Among these deep learning techniques, segmentation methods have particularly evolved, enabling the automatic delineation of anatomical structures such as brain tumors and cardiac tissues, providing critical information for treatment planning [5]. These segmentation models now extend to automated measurements, which are essential for diagnosis and treatment planning [6]. For example, volumetric analysis of organs and precise distance calculations between anatomical landmarks have become integral to personalized medicine, reducing human error and increasing reproducibility [7]. Recent approaches to automated eyeball volume measurement, such as the method proposed by [8], leverage neural network-based segmentation to improve estimation accuracy.

To ophthalmologic and neurologic physicians, precise measurements of eyeball volume, optic nerve sheath diameter, and apex-to-eyeball distance are critical. Eyeball volume is essential for diagnosing and managing conditions such as enophthalmos and proptosis, where the position and size of the eyeball affect both function and aesthetics [8]. Measurement of eyeball volume is instrumental in understanding ocular anatomy and its correlation with various refractive errors such as myopia or hyperopia. Variations in eyeball volume can serve as indicators of pathological conditions, such as glaucoma, where intraocular pressure changes may influence ocular morphology. For orbital surgery, such as reduction of orbital wall fracture, orbital tumor removal, eyeball-orbit ratio information is essential for safe surgery. The optic nerve sheath diameter is a crucial indicator for detecting elevated intracranial pressure, with important implications for both neurologists and critical care physicians [9,10]. Changes in ONSD over time can indicate disease progression or response to treatment, making it a useful parameter in the management of intracranial pathologic conditions such as stroke, traumatic brain injury, and encephalopathy. The increasing diameter can reflect occurrence of optic nerve sheath tumor like meningioma, which might cause blindness to the patients. The apex-to-eyeball distance is vital for surgical planning and avoiding complications during procedures involving the orbit and optic nerve [11]. The AED can aid in evaluating conditions that may involve the optic nerve, such as optic neuritis, orbital tumors, and compressive lesions. The significance of these parameters extends beyond ophthalmology, as it can be relevant in neurology, endocrinology, and even in trauma assessments where orbital involvement is suspected. The automation of these measurements enhances both the accuracy and efficiency of clinical and surgical practice, thereby improving patient outcome, reducing surgical complications and finally advancing ocular research.

Current studies demonstrate the potential of AI techniques in accurately measuring these metrics. For instance, deep neural networks have been applied to estimate the volume of the left ventricle from MRI images, demonstrating the feasibility of AI-based volumetric analysis [12]. Litjens et al. [1] conducted a comprehensive survey on deep learning in medical image analysis, showcasing its application in various diagnostic processes. Shen et al. [3] highlighted the effectiveness of deep learning in improving the accuracy of medical imaging tasks, including segmentation and classification. Maissan et al. [5] specifically addressed the measurement of optic nerve sheath diameter using ultrasonography, underscoring the need for precise automated methods. Despite these advancements, there remains a significant demand for more refined and automated techniques to ensure consistency and accuracy in clinical practice.

This paper proposes automated measurements in oculoplastic surgery by developing a segmentation model using Attention U-Net [13] combined with positional encoding to handle 3D data efficiently [14]. The model automatically segments the eyeball and optic nerve in orbital CT scans, enabling the calculation of volume and distance measurements.

The key contributions of this research are followed. First, by incorporating positional encoding, the model improves segmentation accuracy for CT images by better contextualizing slice positions, which reduces reliance on high-performance computing resources. This approach enables efficient processing of 3D medical data as 2D images, making it broadly applicable to various medical modalities, such as CT and MRI, while also facilitating its use in settings with limited computational resources. Second, the proposed method automates the calculation of volume, diameter, and distance based on segmentation results. This automation has the potential to streamline the diagnostic process, standardize measurements, and improve clinical workflow efficiency. Finally, the deep learning results for structures with high segmentation performance, such as the eyeball, were validated through multiple approaches, including comparisons with manual measurements by ophthalmology experts, demonstrating accuracy and reliability for eyeball volume measurement. In contrast, automated measurements for the optic nerve revealed limitations due to suboptimal segmentation performance.

## 2. Materials and Methods

### 2.1. Data Preparation

The present work was performed with the approval of the Institutional Review Board (IRB) of the Catholic Medical Center, and the privacy, confidentiality, and security of the health information were protected. The orbital CT data were collected (from de-identified human subjects) from hospitals affiliated with the College of Medicine, the Catholic University of Korea (St. Vincent Hospital and Yeouido St. Mary’s Hospital). The orbital CT scans were obtained from January 2016 to December 2020. The dataset contained 78 orbital CT scans from Korean men and women ranging in age from 12 years to 80 years. According to the ICH (International Council for Harmonisation) guidelines, age groups are categorized as follows: children (24 months–11 years), adolescents (12–18 years), adults (19–64 years), and elderly (65-years) [15]. Since there were insufficient data for children, this age group was excluded from the analysis. To further detail the dataset, the characteristics were summarized in Table 1.

To obtain the measurements of eyeball volume, optic nerve sheath diameter, and apex-to-eyeball distance, segmentation of the eyeball and optic nerve was necessary. Four surgeons conducted the masking for the segmentation. Once the CT scans were masked to each corresponding orbital region, the CT slice and label slice were cropped to focus on the relevant voxels. The left and right orbits were subsequently separated to address data scarcity by doubling the available data. Finally, axial view slices were then generated from these cropped regions.

### 2.2. Segmentation Model

#### 2.2.1. Literature Review on U-Net Based Segmentation Models in Medical Imaging

Deep learning segmentation models, particularly those based on U-Net architectures, have become essential in medical image analysis, offering robust solutions for complex segmentation tasks [16]. Traditional 3D U-Nets, which operate on volumetric (voxel) data, are frequently applied for the segmentation of CT and MRI scans [17], but they come with significant demands on computational resources, as noted by Çiçek et al. [18]. These models require extensive memory and processing power, making them challenging to implement in resource-limited settings.

Milletari et al. [19] introduced the V-Net, designed specifically for prostate MRI segmentation. Although effective, V-Net is sensitive to intensity variations across slices, which can lead to inconsistencies in segmentation quality. Other approaches, such as the Multi-scale 3D CNN proposed by Kamnitsas et al. [20], aimed to improve segmentation accuracy by incorporating multi-scale information, yet faced challenges with high memory usage and prolonged training times.

To address some of these limitations, Zhou et al. [21] developed U-Net++, an enhanced U-Net architecture that reduces skip connections and is less prone to overfitting. However, its performance can still be affected by small datasets, limiting its generalizability. The nnU-Net by Isensee et al. [22] further automated the preprocessing and hyperparameter optimization for segmentation across diverse medical imaging modalities, achieving high accuracy but requiring extensive data preprocessing.

In the context of orbital CT segmentation, Chung et al. [23] applied SEQ-UNET, which demonstrated moderate accuracy but had large dataset requirements to maintain performance. Sungho et al. [24] proposed an augmented SEQ-UNET with data augmentation (DA) to improve segmentation of small tumors, although its effectiveness remains limited for smaller structures.

Table 2 provides a summary of these recent segmentation techniques, outlining key performance metrics such as the Dice score, and highlighting the computational and practical limitations of each model. This overview clarifies how our study builds upon previous work, addressing specific gaps and challenges in orbital CT segmentation through the integration of Attention U-Net with positional encoding.

#### 2.2.2. The Proposed Segmentation Model

For segmentation, the Attention U-Net was utilized as the base model. Unlike previous segmentation models that often require high computational resources or struggle with spatial inconsistencies, as shown in Table 2, our approach preserves spatial context across slices by integrating positional encoding, thereby improving segmentation accuracy for complex anatomical structures like the eyeball and optic nerve while also being computationally efficient. While 3D data such as CT scans typically employ 3D-Unet models, which segment based on voxel data [18], we chose a 2D-Unet based approach to ensure applicability in hospitals without high-performance computing resources, as discussed in [23]. This approach helps maintain the three-dimensional characteristics of the data without requiring extensive computational power, aligning with the sequential U-Net approach highlighted in [24]. To preserve the three-dimensional context while processing 2D slices, we incorporated slice position information into the Attention U-Net input using positional encoding. This enhancement allows the model to retain spatial information from the entire volume. The method of adding positional encoding to the Attention U-Net input is illustrated in Figure 1.

### 2.3. Method of Measuring Volume, Diameter, and Distance

(A) Eyeball volume measurement method: The fundamental method for measuring eyeball volume involves multiplying the area corresponding to the eyeball in each slice by the voxel volume. The ground truth volume is calculated using this method by applying the formula to the eyeball masking results from the slices. The predicted volume is obtained by applying the same formula to the segmentation results of the Attention U-Net with Positional Encoding. The ophthalmologists’ results are derived by manually delineating the eyeball region on each slice using the PAX system and then applying the volume calculation formula to the segmentation results. These results will be compared in the Results Section
(1)$$\mathrm{Volume}=\sum_{i=1}^{n}\left({\mathrm{Area}}_{i}\times \mathrm{Slice}\;\mathrm{Thickness}\right)$$

In the above equation, ${\mathrm{Area}}_{i}$ represents the area of the eyeball in the *i*-th slice, and Slice Thickness is the thickness of each CT slice. Based on (Equation 1), ground truth, predicted, and ophthalmologist’s measurement are the following. Slice Thickness is 3mm in both of St. Vincent Hospital and Yeouido St. Mary’s Hospital.

Ground truth: Apply the volume calculation formula (Equation 1) to the manual masking results from each slice.Predicted: Apply the volume calculation formula (Equation 1) to the segmentation results of the Attention U-Net with Positional Encoding.Ophthalmologist’s: Utilize the PAX system to manually delineate the eyeball region in each slice and applying the volume calculation formula to these results.

(B) Optic Nerve Sheath Diameter (ONSD) measurement method: The method for measuring the ONSD involves selecting the first and second slices where the optic nerve class appears from the model’s segmentation results and measuring the diameter on that slice. The ground truth diameter is obtained by applying the measurement to the masking results from the selected slice. These results will be compared in the Results Section.
(2)$${\mathrm{ONSD}}_{eye,i}=\frac{1}{2}\left(D_{\mathrm{eye},\;\mathrm{longest}}+D_{\mathrm{eye},\;\mathrm{shortest}}\right)$$

In the above equation, ${\mathrm{ONSD}}_{eye,i}$ represents the mean of the eye’s longest and shortest diameter of the optic nerve. Here, eye indicates whether it is the right or left eye, while *i* indicates the slice number. To measure the ONSD, we retrieve the first and second slices of the segmentation results, so *i* can be either 1 or 2.

Based on (Equation 2), the ground truth and predicted measurements are as follows:Ground truth: Measure the mean diameter on the first and second slices where the optic nerve appears from the manual masking results.Predicted: Measure the mean diameter on the first two slices where the optic nerve is detected from the segmentation results of the Attention U-Net with Positional Encoding.

(C) Apex-to-Eyeball Distance (AED) measurement method: The method for measuring the apex-to-eyeball distance involves calculating the Euclidean distance based on the central points of the optic nerve region in the starting and ending slices, as viewed in the axial plane. The segmentation results from the model and manual masking are used to determine these points. The ground truth distance is obtained by applying the calculation to the manual masking results. The predicted distance is calculated from the segmentation results of the model used in our segmentation task. These results will also be compared in the Results Section.
(3)$$\mathrm{AED}=\sqrt{{\left(x_{2}-x_{1}\right)}^{2}+{\left(y_{2}-y_{1}\right)}^{2}+{\left(n\times \mathrm{Slice}\;\mathrm{Thickness}\right)}^{2}}$$

In Equation (Equation 3), $\left(x_{1},y_{1}\right)$ and $\left(x_{2},y_{2}\right)$ represent the central points of the optic nerve region in the starting and ending slices, respectively. Here, *n* is the number of slices between the start and end slices, and Slice Thickness is the thickness of each CT slice (3 mm).

Based on (Equation 3), the ground truth and predicted measurements are as follows:Ground truth: Calculate the Euclidean distance using the central points from the manual masking results on the starting and ending slices.Predicted: Calculate the Euclidean distance using the central points from the segmentation results of the Attention U-Net with Positional Encoding on the starting and ending slices.

## 3. Results

### 3.1. Segmentation of Eyeball and Optic Nerve

The segmentation performance of the proposed segmentation model was evaluated using following metrics: Dice Score, Intersection over Union (IoU), sensitivity, and specificity. We employed 5-fold cross-validation, and the overall performance metrics for the segmentation of the eyeball, optic nerve, and background are summarized in Table 3. As summarized in Table 3, positional encoding improved segmentation accuracy by incorporating the slice position information, particularly for the optic nerve.

As seen in Table 3, positional encoding leads to an improvement in both Dice score and IoU, especially in challenging regions such as the optic nerve. This result reflects the importance of incorporating slice position information when processing 3D medical images into 2D slices, with positional encoding effectively providing this slice order information.

Additionally, the Dice score for each segmentation class is provided in Table 4. The Dice score for eyeball segmentation in our model was significantly higher than that of other approaches, such as [23], demonstrating improved performance in segmenting orbital structures. These results indicate that the method performs well in segmenting the background and the eyeball but shows lower performance for the optic nerve, even when using positional encoding.

To further illustrate the segmentation results, Figure 2 presents an example of segmentation result. Each example includes the original CT slice, the ground truth, and the segmentation result generated by our method. The first two rows show segmentation results for the eyeball, whereas the last two rows illustrate the segmentation of the optic nerve. The results demonstrate the ability of the proposed method to accurately segment the eyeball and optic nerve structures from CT images, although the optic nerve segmentation remains challenging, as reflected in the Dice score.

### 3.2. Eyeball Volume

#### 3.2.1. Results of Eyeball Volume Measurements

Table 5 presents the comparison of eyeball volumes measured using ’Ground Truth’ and ’Predicted’ for both male and female CT data across different age groups.

For males aged 19–65, the ’Ground Truth’ volumes were 8.89 ± 2.37 cm^3^ (right) and 8.66 ± 2.48 cm^3^ (left), while the ’Predicted’ volumes were 9.75 ± 2.45 cm^3^ and 9.35 ± 2.74 cm^3^. Similarly, for males over 66, the ’Ground Truth’ was 7.50 ± 3.18 cm^3^ (right) and 7.50 ± 3.01 cm^3^ (left), with ’Predicted’ values were 8.36 ± 2.77 cm^3^ (right) and 8.16 ± 2.29 cm^3^ (left), respectively.

In both male age groups, the predicted volumes were slightly larger than the ground truth on average. The root mean square error (RMSE) values ranged from 1.27 to 1.61 cm^3^, while the mean absolute percentage error (MAPE) remained relatively low, between 0.11 and 0.17. This suggests that, although the predicted volumes are consistently larger than the ground truth, the error remains within a reasonable range. In other words, the focus shifts to the variability and absolute differences, highlighting that the segmentation model tends to overestimate the eyeball volumes slightly, but the overall discrepancies are not substantial.

For females aged 19-65 years, the ’Ground Truth’ measurements were 6.95 ± 1.08 cm^3^ (right) and 6.76 ± 1.21 cm^3^ (left), while the ’Predicted’ volumes were 8.35 ± 1.68 cm^3^ (right) and 7.96 ± 1.83 cm^3^ (left). For the female group 66 or older, the ’Ground Truth’ was 6.86 ± 1.32 cm^3^ (right) and 6.52 ± 1.67 cm^3^ (left), respectively, with the measurements of ’Predicted’ of 8.09 ± 1.93 cm^3^ (right) and 7.78 ± 2.29 cm^3^ (left).

The female group showed higher RMSE values compared to the male group, ranging from 1.07 to 2.02 cm^3^. The MAPE values were also higher, particularly in younger females, reaching 0.23. This indicates that the segmentation model struggled more with predicting female eyeball volumes, leading to larger discrepancies between the predicted and ground truth volumes. The larger errors in the female group suggest a potential limitation in the model’s performance for this demographic, where the predicted volumes consistently deviate more from the measurements based on doctor’s annotation.

For the “sorted” group, data points with relative errors exceeding 0.35 for both the right and left eyes were excluded. After removing six high-error cases, the ’Ground Truth’ for both the right and left eyeballs were 6.99 ± 1.13 cm^3^. The ’Predicted volumes were 7.89 ± 1.54 cm^3^ (right) and 7.46 ± 1.66 cm^3^ (left), with RMSE values of 1.25 cm^3^ (right) and 1.07 cm^3^ (left), respectively. Removing high-error outliers enabled a more detailed analysis, providing better insight into the model’s performance in eyeball volume predictions.

#### 3.2.2. Evaluation of Eyeball Volume Measurements: Compared to ’Ground Truth’

To evaluate the similarity between the predicted segmentation results and the ground truth, we applied the Kolmogorov-Smirnov Test (KS Test) [25] and Mann-Whitney U Test (MW Test) [26]. The KS Test is a non-parametric method used to compare two distributions and assess if their differences are statistically significant. We employed the two-sample KS Test to examine whether distributions of ’Ground Truth’ and ’Predicted’ are similar. Meanwhile, the MW Test compares the medians of two independent groups to evaluate whether their distributions differ significantly. We opted for the MW Test due to the small sample size and the low tendency for the data to follow a normal distribution. Both tests provide *p*-values, where a larger *p*-value indicates no significant difference between the distributions, implying that the model’s predictions closely match the ground truth.

The results of both tests (shown in Table 6) demonstrate that, for the majority of the data, ’Predicted’ for both males and females show no significant difference from ’Ground Truth’, with most *p*-values exceeding 0.05. This indicates that the segmentation model generally performs well in estimating eyeball volumes across different age groups and genders. However, some discrepancies were observed, particularly in the female group aged 19–64.

In the case of the female group aged 19–64, the KS and MW tests both showed statistically significant differences between the predicted and ground truth volumes (*p*-values < 0.05), indicating that the model’s segmentation accuracy for this group was not as reliable. This highlights a potential limitation of the model for this demographic, where segmentation results deviate from the actual measurements.

To address the issue of high relative errors in the female group aged 19–64, six data points with relative errors exceeding 0.35 for both eyes were removed. This was necessary due to the small dataset size, as large errors could distort the overall interpretation of the model’s performance. After removing these outliers, the KS and MW test results showed improved *p*-values, indicating that the distributions of predicted and ground truth values became more aligned. This adjustment enhanced the reliability of the segmentation model’s performance and provided a clearer interpretation of the results. In the Discussion Section, we will further illustrate the similarity between the two distributions through visualized methods, providing deeper insights into the model’s performance.

#### 3.2.3. Evaluation of Eyeball Volume Measurements: Compared to ‘Ophthalmologist’s’

This study compares eyeball volumes measured using model’s segmentation results based on a deep learning model(‘Prediced’), and direct measurements taken by ophthalmologists using the PAX system(Ophthalmologist’s). For this analysis, the data consist of 16 samples in the 19–64 age group, including 8 males and 8 females. Table 7 compares ’Predicted’ and ’Ophthalmologist’s’ to explore the average eyeball volume and the standard deviation for both methods, as well as the root mean square error (RMSE) and the mean absolute percentage error (MAPE).

For male data, the ophthalmologist’s measurements of the eyeball volumes were 9.60 ± 1.55 cm^3^ (right) and 9.56 ± 1.24 cm^3^ (left). The predicted volumes were slightly lower, at 8.41 ± 1.53 cm^3^ (right) and 8.14 ± 1.47 cm^3^ (left). The RMSE values for the right and left eyes were 1.84 cm^3^ and 1.97 cm^3^, respectively, with MAPE values of 0.16 and 0.17. Although the predicted values tend to slightly underestimate the ophthalmologist’s measurements, the overall error remains within a reasonable range, suggesting a fairly accurate performance of the segmentation model.

For female data, the ophthalmologist’s measurements were 9.48 ± 1.49 cm^3^ (right) and 9.49 ± 1.46 cm^3^ (left). Similarly to the male group, the predicted volumes were slightly lower at 8.61 ± 1.47 cm^3^ (right) and 8.14 ± 1.72 cm^3^ (left). The RMSE values were 1.66 cm^3^ for the right eye and 2.15 cm^3^ for the left eye, with MAPE values of 0.15 and 0.18, respectively. This suggests that the segmentation model performed slightly better for the right eye but showed larger discrepancies for the left eye, especially in the female group.

Table 8 presents the results of KS and MW Test of Ophthalmologist’s and prediced volume.

For male data, the KS Test *p*-values for both the right and left eyes were 0.28, indicating no statistically significant difference between the predicted and ophthalmologist-measured distributions. The MW Test also showed non-significant results, with *p*-values of 0.13 for both eyes. These results suggest that the segmentation model’s predictions are statistically comparable to the ophthalmologist’s measurements for male subjects, with no significant deviation in the distribution of values.

For female subjects, the KS Test *p*-values were 0.66 for the right eye and 0.28 for the left eye, indicating no significant difference in the distribution of predicted and measured values. Similarly, the MW Test results showed *p*-values of 0.44 for the right eye and 0.16 for the left eye, further confirming that the predicted and ophthalmologist-measured volumes for females are statistically similar. These results suggest that the segmentation model performs consistently across genders, without significant statistical discrepancies between predicted and the ophthalmologist’s measurements.

#### 3.2.4. Gender and Age-Specific Trends in Predicted Eyeball Volumes

Analyzing the trends in eyeball volumes across different age groups and genders, based on Figure 3 and Figure 4—where the mean is represented by bars and the standard deviation by error bars—we observe the following:Male Group: Both right and left eyeball volumes show an increase in the 19–64 age group compared to those aged 65+. The predicted segmentation volumes closely follow the ground truth trends, but with a tendency to slightly overestimate in both age groups. The variability, as indicated by the error bars, remains relatively high across all age groups for males, reflecting some inconsistency in the segmentation predictions.Female Group: A similar trend is observed in the 19-64 age group, with reduced variability in the sorted data, providing more consistent estimates between the predicted and ground truth values. This suggests that filtering out high-error data points helps improve prediction accuracy. However, for the 65+ group, the segmentation results still show a tendency to slightly overestimate eyeball volumes, similar to the male group.Overall Trend: Across both genders, the predicted segmentation results align well with the ground truth measurements. However, the model shows a tendency to slightly overestimate eyeball volumes in older age groups (65+), particularly in males. For females, especially in the sorted dataset, the alignment between predicted and ground truth values improves, reducing discrepancies.Variability: Variability in the predicted eyeball volumes is generally higher for males. For females, especially in the sorted 19–64 group, trends are more consistent for predicted values. This indicates that the sorting process helps in minimizing the deviations between predicted and ground truth volumes for females.

### 3.3. Optic Nerve Sheath Diameter (ONSD)

The analysis of ONSD measurements based on ‘Ground Truth’ and ‘Predicted’ shows significant discrepancies between the predicted and actual values, particularly for older individuals and female patients. As shown in Table 9 and Table 10, segmentation results for the optic nerve were less accurate, as reflected in the lower Dice score (0.6387 ± 0.1394) compared to the eyeball (0.8466 ± 0.1154). This reduced accuracy is further evidenced by the larger differences in ONSD measurements and lower *p*-values in the Kolmogorov-Smirnov (KS) test, especially for the second measurement location across all groups.

The “first” measurement location refers to the axial view slice where the optic nerve is first visible, while the “second” location is the next consecutive slice, taken 3 mm apart, in the axial view. For males aged 19–64, the model’s predicted ONSD values closely matched the ground truth at the first location, but significant differences were observed at the second location, with the KS *p*-value resulting below 0.01. Older males (65+) also showed a similar pattern, with greater deviations at the second location. Females aged 19–64 followed a similar trend as the 19–64 aged male group, also showing KS test *p*-values below 0.01. The model’s performance for older females (65+) was notably poorer, with large deviations in predicted values from the actual measurements.

Overall, the segmentation model struggled with accurately estimating ONSD, especially in older individuals and females, highlighting a need for improved segmentation performance for the optic nerve to enhance the reliability of ONSD measurements in clinical applications.

### 3.4. Apex-to-Eyeball Distance (AED)

Table 11 shows the comparison of AED measured using ground truth and the predicted segmentation results. The analysis reveals differences between two measurements across different age groups and genders.

For males aged 19–64, the ground truth measurements were 2.26 ± 0.85 cm (right) and 2.62 ± 0.90 cm (left), while the predicted values were significantly lower at 1.63 ± 0.67 cm (right) and 1.73 ± 0.61 cm (left). This indicates a tendency of the model to underestimate the distance in this age group, with the KS test yielding *p*-values less than 0.01, showing significant differences between the distributions of actual and predicted values. In older males (65+), the discrepancy was smaller, with ground truth values of 2.30 ± 1.02 cm (right) and 2.30 ± 0.89 cm (left), and predicted values of 1.75 ± 0.60 cm (right) and 1.76 ± 0.58 cm (left). However, the KS test for the left eye in this group yielded a *p*-value of 0.5459, suggesting that the predictions align better with the ground truth for this particular subset.

For females aged 19–64, the predicted values were slightly lower compared to the ground truth. The measured distances were 2.12 ± 0.89 cm (right) and 2.21 ± 0.65 cm (left) for the ground truth, while the predicted values were 1.70 ± 0.65 cm (right) and 1.70 ± 0.04 cm (left). Both the right and left eyes had significant differences, with *p*-values less than 0.01 in the KS test, indicating a statistically significant mismatch between predicted and actual measurements. For older females (65+), the ground truth values were 2.33 ± 0.80 cm (right) and 2.32 ± 0.77 cm (left), while the predicted values were slightly lower at 1.68 ± 0.51 cm (right) and 1.66 ± 0.49 cm (left). The KS test for the left eye showed a *p*-value of 0.0904, suggesting a moderate alignment between the predicted and actual values in this age group.

Overall, the analysis indicates that while the model’s segmentation method provides reasonably close predictions for some age and gender groups, it tends to underestimate the apex-to-eyeball distance, especially in males aged 19–64 and females in both age groups. This underestimation highlights the need for further refinement of the model to improve its performance in predicting this metric across different demographics.

## 4. Discussion

First, we will now visualize the distribution of eyeball volume measurement results compared to the ground truth (with sorted data for the female group). The first visualization method we employed is the QQ plot, which helps us assess how well the segmentation values match the ground truth by comparing their distributions. The results were shown in Figure 5.

The QQ plots (Figure 5) for both male and female groups (sorted data for females) show that the predicted segmentation values generally align well with the ground truth. For the male group, the QQ plots illustrate a strong correlation between predicted and actual values, with only minor deviations at the extremes. For the female group (sorted data), the alignment improved significantly after removing the high-error data points, leading to better correspondence between the distributions.

To further assess the relationship between segmentation results and ground truth, we computed the correlation coefficients:Male Right Eye: 0.92Male Left Eye: 0.92Female Right Eye (sorted): 0.87 (previously 0.63 before sorting)Female Left Eye (sorted): 0.87 (previously 0.79 before sorting)

The correlation values indicate strong positive correlations between the segmentation results and the ground truth for both genders, with particularly high values for the male group. For the female group, the correlation improved substantially after sorting, demonstrating that the removal of high-error data cases helped bring the segmentation results closer to the ground truth.

The Bland-Altman plots highlight the agreement between the predicted segmentation results and ground truth for eyeball volume measurements. Across both male and female groups, the model generally shows a slight overestimation of volumes, with most differences falling within the upper and lower limits of agreement as shown in Figure 6.

For the male group, the mean differences are 0.86 (right) and 0.69 (left), indicating a relatively close match between the predicted and actual values. Some deviation is observed at higher volume values, but overall, the predictions are consistent with the ground truth. In the female group, the mean differences are slightly larger, particularly for the right eye (0.90), while the left eye shows a smaller mean difference (0.68). The predictions for the female group display slightly more variance, especially at higher volumes, but remain within the acceptable range. While the segmentation model tends to overestimate slightly, the differences are generally within reasonable limits, and the model performs adequately across different eyes and genders.

The histograms and Kernel Density Estimation (KDE) plots in Figure 7 provide a comparison between the predicted segmentation results and ground truth values for the eyeball volumes. Overall, the predicted values tend to slightly overestimate the actual volumes, particularly for the male right eye, as shown by the skew in the histograms and earlier peaks in the KDE plots. However, the left eye for both genders shows a closer match between predicted and actual values, with the KDE curves more aligned. For the female group, after sorting, the distributions are notably improved, showing better agreement between the predicted and actual values. This suggests that removing high-error data points helped improve the model’s prediction accuracy.

The comparison of eyeball volume measurements demonstrates the accuracy and reliability of our proposed segmentation method. Prior research has reported varying degrees of success in accurately segmenting the eyeball, with Dice scores ranging from 0.75 to 0.85. Our method achieved a Dice score of 0.8466, placing it within the upper range of existing literature. This consistency with prior findings validates our approach and suggests that our method can be reliably used for clinical and research purposes. Furthermore, the specific improvements in sensitivity and specificity metrics highlight the robustness of our segmentation technique in distinguishing between the eyeball, optic nerve, and background.

In the evaluation of optic nerve sheath diameter (ONSD) measurements, our results do not align with previous studies in terms of accuracy. While earlier research reported mean differences from ground truth values within ±0.2 mm, the present study shows more significant deviations. The model’s segmentation accuracy for ONSD, as indicated by a Dice score of 0.6387, reflects limitations in its ability to precisely capture the optic nerve sheath. This reduced accuracy highlights the need for further refinement of the segmentation approach, particularly given the clinical importance of ONSD measurements for diagnosing and monitoring conditions like elevated intracranial pressure. Accurate ONSD measurements are critical for clinical decision-making, and while our method demonstrates potential, it requires additional improvement to be considered reliable for clinical application.

Regarding the apex-to-eyeball distance (AED) measurements, the results also indicate notable discrepancies between predicted and actual values. Contrary to the initial assumption of high accuracy, the *p*-values from the Kolmogorov-Smirnov test reveal statistically significant differences, particularly in certain age and gender groups, such as males aged 19–64 and females across both age ranges. The model tends to underestimate AED, and while the overall trends are consistent, the magnitude of the error suggests that the current segmentation approach needs enhancement. AED is crucial for various ophthalmologic and neurologic evaluations, and although our method shows promise, the observed deviations point to the necessity of further optimization to ensure dependable measurements across diverse populations. The variability in results across different demographic subsets further underscores the need for tailored adjustments in the model to enhance its robustness and precision.

In an effort to improve the segmentation accuracy of the optic nerve due to the limited number of data samples, we experimented with both denoising diffusion probabilistic models (DDPMs) [27] for oversampling using generative models [28,29] and transfer learning using the relatively larger Liver Tumor Segmentation Benchmark dataset (LiTS) as the source dataset [24]. These approaches improved the dice score to some extent, especially with transition learning, where the optic nerve segmentation score varied between 0.66 and 0.75 with transfer learning, but these improvements did not significantly improve the accuracy of the automated measurements. The small size of the optic nerve and its relatively limited representation in the dataset pose inherent challenges to robust segmentation. Therefore, increasing the size and diversity of datasets is necessary to achieve clinically reliable segmentation and robust automated measurements for complex anatomical regions and remains future work.

## 5. Conclusions

This study demonstrates the significant achievements of the proposed Attention U-Net with Positional Encoding in accurately segmenting orbital structures, particularly the eyeball. The model consistently achieved high Dice scores and low error rates for eyeball volume measurements across various age and gender groups, highlighting its clinical relevance and potential for automating routine ocular assessments. By incorporating positional encoding, the segmentation accuracy was notably improved, especially in complex regions like the optic nerve.This enhancement underscores the model’s potential for clinical relevance in automating ocular assessments, although further refinement is required for structures such as the optic nerve.

While the model exhibited high performance in eyeball segmentation, providing reliable and accurate volume estimates, its performance in optic nerve sheath diameter (ONSD) and apex-to-eyeball distance (AED) measurements was less robust. The optic nerve, being a more challenging structure with its narrow and elongated shape, presented difficulties in achieving the same level of segmentation precision. However, the results do indicate the potential of the model, as demonstrated by its capability to approximate these measurements and its promise for further improvement.

Looking forward, enhancing the segmentation accuracy for finer structures like the optic nerve will require both technical advancements and the expansion of training datasets. By incorporating additional techniques specifically designed to capture narrow and elongated structures, and by increasing the diversity and quantity of training data, we aim to further improve the reliability of ONSD and AED measurements. Despite current challenges, the results validate the potential of deep learning-based segmentation in automating the measurement of orbital structures, and with continued development, this approach could become a reliable tool for clinical applications in the near future.

## Figures and Tables

**Figure 1 diagnostics-14-02643-f001:**
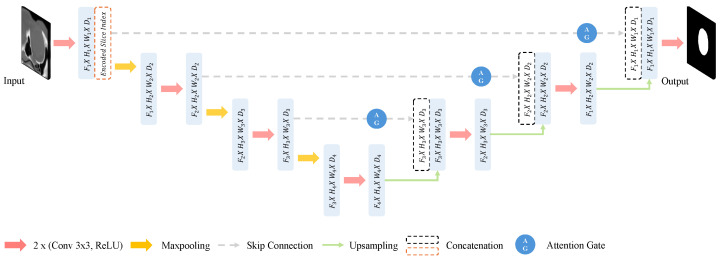
Incorporation of positional encoding into Attention U-Net input to retain 3D context in 2D slice processing.

**Figure 2 diagnostics-14-02643-f002:**
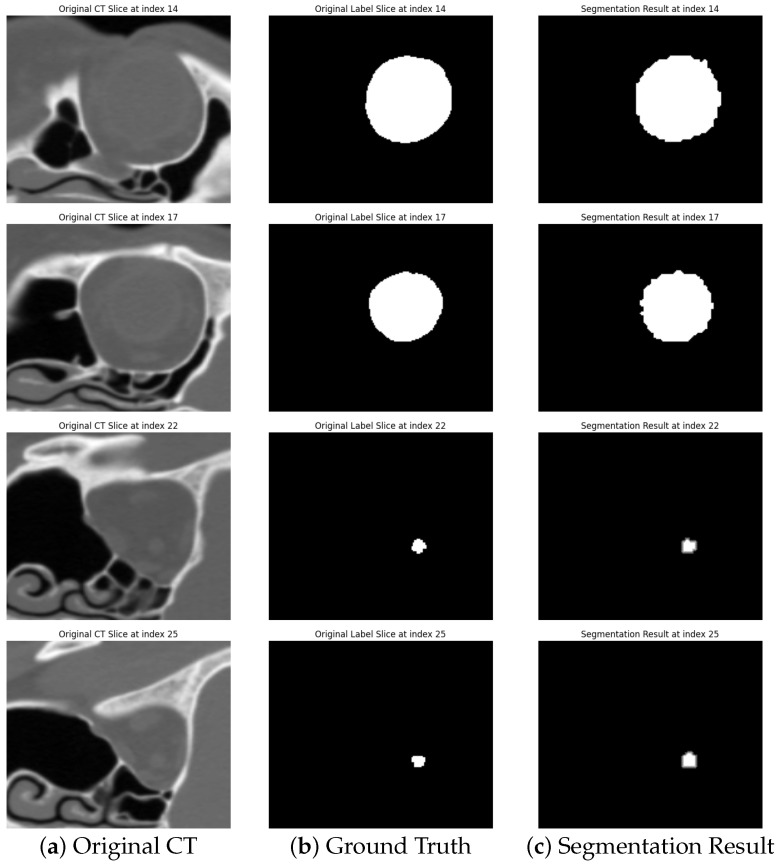
Examples of segmentation results.

**Figure 3 diagnostics-14-02643-f003:**
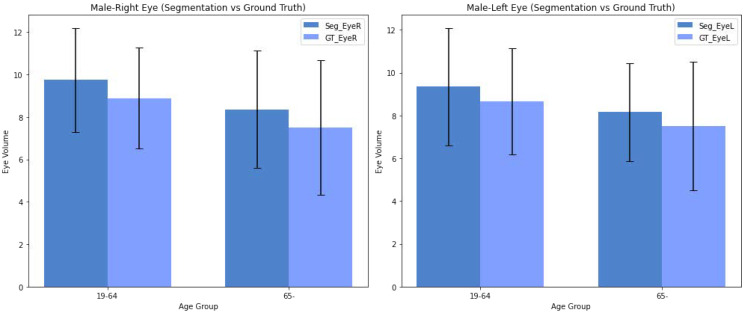
Eyeball Volumes of male for Ground Truth and Predicted Values, with an Analysis of Age- and Gender-Specific Trends.

**Figure 4 diagnostics-14-02643-f004:**
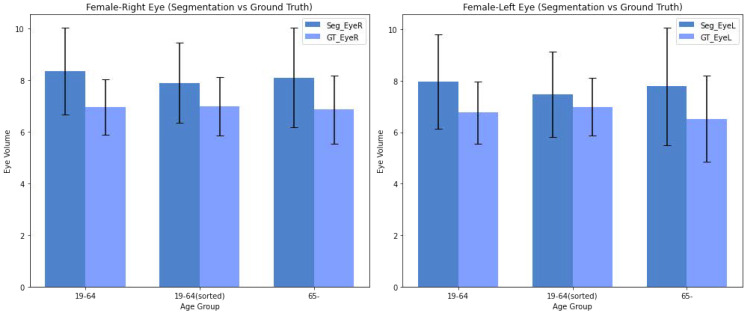
Eyeball Volumes of female for Ground Truth and Predicted Values, with an Analysis of Age- and Gender-Specific Trends.

**Figure 5 diagnostics-14-02643-f005:**
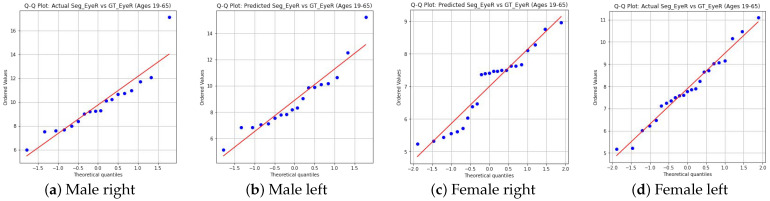
QQ Plots: Predicted vs. Actual Segmentation vs. Ground Truth for Male and Female Groups (Ages 19–65).

**Figure 6 diagnostics-14-02643-f006:**
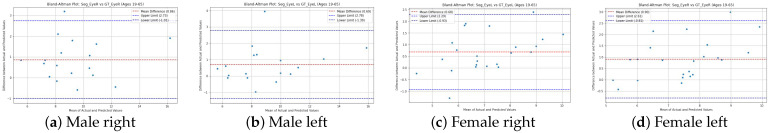
Bland-Altman plots: Predicted vs. Actual Segmentation vs. Ground Truth for Male and Female Groups (Ages 19–65).

**Figure 7 diagnostics-14-02643-f007:**
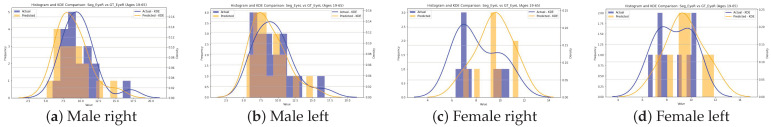
Histogram and KDE: Predicted vs. Actual Segmentation vs. Ground Truth for Male and Female Groups (Ages 19–65).

**Table 1 diagnostics-14-02643-t001:** Characteristics of the orbital CT dataset by gender and age group.

Gender	Age Group	Number of Subjects
	12–18	4
Male (29)	19–64	18
	65–	7
	12–18	3
Female (45)	19–64	29
	65–	13

**Table 2 diagnostics-14-02643-t002:** Summary of Related Works on Medical Image Segmentation.

Reference	Dataset	Model	Results (Dice Score)	Limitations
Cicek [18] 2016	Kidney and Liver CT	3D U-Net	0.89	High computational resource requirements
Milletari [19] 2016	Prostate MRI	V-Net	0.86	Affected by intensity variations across slices
Kamnitsas [20] 2017	Brain MRI	Multi-scale 3D CNN	0.82	High memory usage and long training
Zhou [21] 2018	Lung CT	U-Net++	0.91	Overfitting on small datasets
Isensee [22] 2021	Medical Segmentation Decathlon	nnU-Net	0.96	Requires extensive data preprocessing
Chung [23] 2022	Orbital CT	SEQ-UNET	0.79	Larger dataset requirements
Sungho [24] 2022	Orbital CT	SEQ-UNET + DA	0.68	Low performance on small tumors

**Table 3 diagnostics-14-02643-t003:** Overall Segmentation Performance with and without Positional Encoding (Mean ± Std).

Metric	With Positional Encoding	Without Positional Encoding
Dice Score	0.7638 ± 0.0658	0.6995 ± 0.0442
IoU	0.5723 ± 0.0954	0.4149 ± 0.1854
Sensitivity	0.5940 ± 0.1128	0.3957 ± 0.1320
Specificity	0.9301 ± 0.0397	0.8675 ± 0.1004

**Table 4 diagnostics-14-02643-t004:** Dice Score for Each Segmentation Class.

Class	Dice Score
Background	0.9821 ± 0.0073
Eyeball	0.8466 ± 0.1154
Optic Nerve	0.6387 ± 0.1394

**Table 5 diagnostics-14-02643-t005:** Eyeball Volume by Age Group and Gender: Comparison between Ground Truth and Predicted.

Gender	AGE	Side	Ground Truth (${\mathrm{cm}}^{3}$)	Predicted (${\mathrm{cm}}^{3}$)	RMSE (${\mathrm{cm}}^{3}$)	MAPE
Male	19–64	Right	8.89 ± 2.37	9.75 ± 2.45	1.28	0.12
		Left	8.66 ± 2.48	9.35 ± 2.74	1.27	0.11
	6–6–	Right	7.50 ± 3.18	8.36 ± 2.77	1.61	0.17
		Left	7.50 ± 3.01	8.16 ± 2.29	1.40	0.16
Female	19–65	Right	6.95 ± 1.08	8.35 ± 1.68	1.90	0.21
		Left	6.76 ± 1.21	7.96 ± 1.83	1.75	0.20
	19–64 (sorted)	Right	6.99 ± 1.13	7.89 ± 1.54	1.25	0.14
		Left	6.99 ± 1.13	7.46 ± 1.66	1.07	0.12
	66–	Right	6.86 ± 1.32	8.09 ± 1.93	1.72	0.19
		Left	6.52 ± 1.67	7.78 ± 2.29	2.02	0.23

**Table 6 diagnostics-14-02643-t006:** Statistical Comparison of Segmentation Results and Ground Truth: Kolmogorov-Smirnov Test and Mann-Whitney U Test.

Gender	AGE	Side	KS Stat	KS *p*-Value	MW Stat	MW *p*-Value
Male	19–64	Right	0.28	0.50	205.00	0.18
		Left	0.22	0.78	190.00	0.38
Male	66–	Right	0.43	0.58	29.00	0.62
		Left	0.43	0.58	31.00	0.46
Female	19–64	Right	0.41	0.01	626.00	0.00
		Left	0.38	0.03	596.00	0.01
Female	19–64	Right	0.35	0.12	354.00	0.05
	(sorted)	Left	0.30	0.24	330.00	0.15
Female	66–	Right	0.54	0.04	121.00	0.06
		Left	0.54	0.04	125.00	0.04

**Table 7 diagnostics-14-02643-t007:** Eyeball Volume of 19–64 Age Group: Comparison between ‘Ophthalmologist’s’ and ‘Predicted’.

Gender	AGE	Side	Ophthalmologist’s (cm^3^)	Predicted (cm^3^)	RMSE (cm^3^)	MAPE
Male	19–64	Right	9.60 ± 1.55	8.41 ± 1.53	1.84	0.16
		Left	9.56 ± 1.24	8.14 ± 1.47	1.97	0.17
Female	19–64	Right	9.48 ± 1.24	8.61 ± 1.47	1.66	0.15
		Left	9.49 ± 1.46	8.14 ± 1.72	2.15	0.18

**Table 8 diagnostics-14-02643-t008:** Statistical Comparison of Segmentation Results and Ophthalmologist’s: Kolmogorov-Smirnov Test and Mann-Whitney U Test.

Gender	AGE	Side	KS Stat	KS *p*-Value	MW Stat	MW *p*-Value
Male	19–64	Right	0.50	0.28	17.00	0.13
		Left	0.50	0.28	17.00	0.13
Female	19–64	Right	0.38	0.66	24.00	0.44
		Left	0.50	0.28	18.00	0.16

**Table 9 diagnostics-14-02643-t009:** Right Eye’s ONSD by Age Group and Gender: Comparison between Ground Truth and Predicted.

Gender	AGE	Ground Truth (cm)		Predicted (cm)		KS *p*-Value	
		**First**	**Second**	**First**	**Second**	**First**	**Second**
Male	19–64	0.12 ± 0.07	0.27 ± 0.11	0.14 ± 0.03	0.14 ± 0.05	0.2256	≤0.01
Male	65–	0.17 ± 0.11	0.28 ± 0.15	0.21 ± 0.14	0.19 ± 0.13	0.1750	≤0.01
Female	19–64	0.14 ± 0.10	0.22 ± 0.11	0.14 ± 0.05	0.14 ± 0.06	0.0176	≤0.01
Female	65–	0.13 ± 0.10	0.22 ± 0.12	0.12 ± 0.05	0.20 ± 0.13	≤0.01	≤0.01

**Table 10 diagnostics-14-02643-t010:** Left Eye’s ONSD by Age Group and Gender: Comparison between Ground Truth and Predicted.

Gender	AGE	Ground Truth (cm)		Predicted (cm)		KS *p*-Value	
		**First**	**Second**	**First**	**Second**	**First**	**Second**
Male	19–64	0.14 ± 0.10	0.27 ± 0.15	0.17 ± 0.09	0.17 ± 0.07	0.1535	≤0.01
Male	65–	0.17 ± 0.13	0.30 ± 0.17	0.20 ± 0.14	0.23 ± 0.14	0.0880	≤0.01
Female	19–64	0.16 ± 0.16	0.29 ± 0.21	0.13 ± 0.05	0.17 ± 0.08	≤0.01	≤0.01
Female	65–	0.15 ± 0.15	0.23 ± 0.15	0.15 ± 0.02	0.13 ± 0.02	0.0271	≤0.01

**Table 11 diagnostics-14-02643-t011:** Apex-to-Eyeball Distance by Age Group and Gender: Comparison between Ground Truth and Predicted.

Gender	AGE	Ground Truth (cm)		Predicted (cm)		KS *p*-Value	
		**Right**	**Left**	**Right**	**Left**	**Right**	**Left**
Male	19–64	2.26 ± 0.85	2.62 ± 0.90	1.63 ± 0.67	1.73 ± 0.61	≤0.01	0.0368
Male	65–	2.30 ± 1.02	2.30 ± 0.89	1.75 ± 0.60	1.76 ± 0.58	≤0.01	0.5459
Female	19–64	2.12 ± 0.89	2.21 ± 0.65	1.70 ± 0.65	1.70 ± 0.04	≤0.01	≤0.01
Female	65–	2.33 ± 0.80	2.32 ± 0.77	1.68 ± 0.51	1.66 ± 0.49	≤0.01	0.0904

## Data Availability

The datasets generated and/or analyzed during the current study are available from the corresponding author upon reasonable request.

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
