# Peer review of "Segmentation-Based Measurement of Orbital Structures: Achievements in Eyeball Volume Estimation and Barriers in Optic Nerve Analysis"

_diagnostics, 2024, doi:10.3390/diagnostics14232643_

Round 1
Reviewer 1 Report
Comments and Suggestions for Authors
The study provides an interesting application of deep learning in the segmentation of orbital structures; however, its clinical relevance and impact are not clearly presented. To enhance the significance of the study, the authors should better explain how automated measurements of eyeball volume or optic nerve sheath diameter contribute to the diagnosis and management of specific diseases, such as glaucoma or optic neuropathy. The current focus on technical performance without strong clinical context limits the broader applicability of the results.
The manuscript lacks clarity regarding the specific diseases or conditions this segmentation model is designed to assist with. The measurements provided (e.g., eyeball volume and optic nerve sheath diameter) are important, but the study does not sufficiently explain which diseases or clinical applications these measurements target. The authors should clearly identify these conditions (e.g., increased intracranial pressure, enophthalmos, optic neuritis) and explain the value of the automated measurements for diagnosis or treatment.
Although the paper provides some technical advancements, such as using the Attention U-Net with positional encoding, the authors do not make it clear how their work advances the field compared to previous studies. It is essential that the authors situate their work within the context of existing research. To do this, the authors should build a comprehensive table that reviews relevant literature, including previous segmentation techniques and their performance metrics (e.g., Dice score, Intersection over Union). This table should help highlight how the current study contributes to the field, either by improving accuracy, reducing computational costs, or enabling new clinical applications.
The segmentation results, particularly for the optic nerve, are not yet robust enough for clinical use, as evidenced by a Dice score of 0.6387. To improve the model’s performance, the authors could consider adopting a Generative Adversarial Network (GAN) approach, which has proven successful in enhancing segmentation accuracy by generating more realistic training data (Ref: Application of generative adversarial networks (GAN) for ophthalmology image domains: a survey, Eye and Vision, 2022). Additionally, expanding the dataset, particularly for the optic nerve, would improve model training and generalization. This step is crucial, as a small dataset limits the robustness and applicability of the model across diverse clinical cases.
The current dataset is relatively small, which is likely a factor contributing to the model's limited performance, especially for complex structures like the optic nerve. The authors should expand their dataset to include more diverse cases or use external datasets for validation. This would increase the reliability of their results. In cases where data collection is challenging, the authors should consider data augmentation or transfer learning to mitigate the dataset size limitations.
Author Response
Comment 1: The study provides an interesting application of deep learning in the segmentation of orbital structures; however, its clinical relevance and impact are not clearly presented. To enhance the significance of the study, the authors should better explain how automated measurements of eyeball volume or optic nerve sheath diameter contribute to the diagnosis and management of specific diseases, such as glaucoma or optic neuropathy. The current focus on technical performance without strong clinical context limits the broader applicability of the results
Response 1: Thank you for pointing this out. We agree with this comment. Therefore, we have revised the manuscript to better explain the clinical relevance of our proposed measurement method. Specifically, we have clarified the importance of automated measurements of eyeball volume, optic nerve sheath diameter (ONSD), and apex-to-eyeball distance (AED) in diagnosing and managing conditions such as enophthalmos, proptosis, and elevated intracranial pressure. We added detailed discussions on how precise measurements of eyeball volume can assist in conditions where ocular positioning is critical for both function and aesthetics, as in enophthalmos and proptosis (page 2, lines 33 – 39). Additionally, we discussed the role of ONSD in monitoring intracranial pressure, which is particularly valuable for neurologists and critical care physicians (page 2, lines 41 – 45). Finally, how AED can aid in evaluating conditions related to the optic nerve and how the automated measurements enhance clinical and surgical practice, not only in ophthalmology but in various fields as well, were mentioned (page 2, lines 47 – 53).
Comment 2: The manuscript lacks clarity regarding the specific diseases or conditions this segmentation model is designed to assist with. The measurements provided (e.g., eyeball volume and optic nerve sheath diameter) are important, but the study does not sufficiently explain which diseases or clinical applications these measurements target. The authors should clearly identify these conditions (e.g., increased intracranial pressure, enophthalmos, optic neuritis) and explain the value of the automated measurements for diagnosis or treatment.
Response 2: We agree with this comment and have revised the manuscript accordingly to emphasize the clinical applications targeted by our segmentation model. We clarified that the model is relevant for assessing specific conditions, including increased intracranial pressure, enophthalmos, proptosis, and optic neuropathy. The manuscript now includes a more explicit discussion on how these measurements support clinical decision-making, particularly in ophthalmology and neurology. For instance, we describe the significance of ONSD for monitoring elevated intracranial pressure and the role of eyeball volume measurements in identifying ocular pathologies such as glaucoma and refractive errors. These changes can be found on page 2, lines 33 – 53.
Comment 3: Although the paper provides some technical advancements, such as using the Attention U-Net with positional encoding, the authors do not make it clear how their work advances the field compared to previous studies. It is essential that the authors situate their work within the context of existing research. To do this, the authors should build a comprehensive table that reviews relevant literature, including previous segmentation techniques and their performance metrics (e.g., Dice score, Intersection over Union). This table should help highlight how the current study contributes to the field, either by improving accuracy, reducing computational costs, or enabling new clinical applications.
Response 3: Thank you for this insightful suggestion. We have revised the manuscript to better situate our work within the context of existing research on deep-learning segmentation models for medical image analysis. Specifically, we have added a comprehensive table (Table 1) that summarizes recent literature on segmentation models used in medical image analysis domain. These changes can be found on page 2, lines 44 – 48. In addition to the literature review, we have also explained in \textbf{OOO} the rationale for using Attention U-Net with positional encoding, compared to models used in previous studies.(page 2, lines 44 – 48)
Comment 4: The segmentation results, particularly for the optic nerve, are not yet robust enough for clinical use, as evidenced by a Dice score of 0.6387. To improve the model’s performance, the authors could consider adopting a Generative Adversarial Network (GAN) approach, which has proven successful in enhancing segmentation accuracy by generating more realistic training data (Ref: Application of generative adversarial networks (GAN) for ophthalmology image domains: a survey, Eye and Vision, 2022). Additionally, expanding the dataset, particularly for the optic nerve, would improve model training and generalization. This step is crucial, as a small dataset limits the robustness and applicability of the model across diverse clinical cases.
Response 4: Thank you for this suggestion. For testing the effectiveness of GANs in segmentation tasks, we explored an alternative generative approach, the Denoising Diffusion Probabilistic Model (DDPM), to improve segmentation performance. However, in our case, DDPM achieved a Dice score of 0.6492 for optic nerve segmentation, which was only marginally higher than those obtained with our current model, the Attention U-Net with positional encoding, which achieved a Dice score of 0.6387. We agree that expanding the dataset, particularly for the optic nerve, could improve robustness and generalizability, and we plan to pursue this approach in future work. (The discussion about this issue is written in line 480-489)
Comment 5: The current dataset is relatively small, which is likely a factor contributing to the model's limited performance, especially for complex structures like the optic nerve. The authors should expand their dataset to include more diverse cases or use external datasets for validation. This would increase the reliability of their results. In cases where data collection is challenging, the authors should consider data augmentation or transfer learning to mitigate the dataset size limitations.
Response 5: Thank you for this valuable suggestion. We addressed the dataset limitations by implementing transfer learning using the Liver Tumor Segmentation (LiTS) dataset, which allowed us to improve the model's performance for optic nerve segmentation. With transfer learning, the dice score of optic nerve segmentation varied from 0.66 to 0.75, demonstrating an enhancement in segmentation accuracy. However, this improvement in segmentation robustness did not translate to a corresponding increase in the accuracy of automated measurements. For structures like the optic nerve, which occupy a relatively small area in the images, even with generative models or transfer learning, there are inherent limitations. These results suggest that overcoming the constraints of small dataset sizes for such fine structures remains challenging and that a substantial increase in dataset size may be necessary to achieve clinically reliable results. (The discussion about this issue is written in line 480-489)

Reviewer 2 Report
Comments and Suggestions for Authors
The paper introduces a novel approach by integrating positional encoding into the Attention U-Net model, which allows for efficient 3D medical image segmentation even in environments with limited computational resources, such as hospitals without high-end hardware. Overall, the combination of technical innovation and practical applicability makes this study a valuable contribution to the field.
However, there are notable limitations, particularly in optic nerve segmentation accuracy and demographic performance discrepancies (e.g., gender and age-related errors). Additionally, the small and non-diverse dataset raises concerns about the model’s generalizability to broader populations.
Author Response
Comment 1: The paper introduces a novel approach by integrating positional encoding into the Attention U-Net model, which allows for efficient 3D medical image segmentation even in environments with limited computational resources, such as hospitals without high-end hardware. Overall, the combination of technical innovation and practical applicability makes this study a valuable contribution to the field.
However, there are notable limitations, particularly in optic nerve segmentation accuracy and demographic performance discrepancies (e.g., gender and age-related errors). Additionally, the small and non-diverse dataset raises concerns about the model’s generalizability to broader populations.
Response 1: Thank you for insightful feedback on its limitations. To address the challenges related to optic nerve segmentation accuracy, we implemented transfer learning using the LiTS dataset, which led to an improvement in the Dice score for optic nerve segmentation, varying from 0.66 to 0.75. Although this approach enhanced segmentation accuracy to some extent, it did not significantly improve the accuracy of automated measurements, as the optic nerve occupies a relatively small area in the images. This suggests that even with transfer learning, there are inherent limitations in addressing such fine structures with a small dataset, and a substantial increase in dataset size may be necessary for clinically robust segmentation.
We also appreciate your concerns regarding the dataset size and diversity. Although the number of subjects in our dataset is relatively small, we have made efforts to ensure demographic diversity within the available sample, with representation across gender and multiple age groups (as shown in Table 1). This study serves as a preliminary exploration, demonstrating the feasibility and potential of our approach. We agree that further increasing the dataset size and including more diverse demographic groups will enhance the model's robustness and generalizability. We plan to address this in future work by collecting additional data with broader demographic representation, which will allow us to conduct a more comprehensive analysis of demographic performance discrepancies. (The discussion about this issue is written in line 480-489)

Round 2
Reviewer 1 Report
Comments and Suggestions for Authors
The authors' response to questions 4 and 5 does not seem appropriate. GAN and Diffusion models are different algorithms, and the validation performed on the Liver Tumor Segmentation dataset is also strange.
Author Response
Comment 1: The authors' response to questions 4 and 5 does not seem appropriate. GAN and Diffusion models are different algorithms, and the validation performed on the Liver Tumor Segmentation dataset is also strange.
Response 1: Thank you for pointing this out. We acknowledge that our initial response did not adequately address your concerns about our choice of diffusion models and the application of the Liver Tumor Segmentation Benchmark dataset (LiTS).
Firstly, we agree that GANs are extensively used for oversampling in scenarios of data imbalance, including eye-related deep learning tasks[1] and beyond [2][3]. However, we opted for diffusion models due to their more stable training dynamics and enhanced capabilities in generating synthetic data. Recent studies support the effectiveness of diffusion models in producing synthetic data that contributes to improved robustness of datasets [4].
Secondly, the LiTS dataset was not employed for validation purposes. It was instead used as a source dataset for transfer learning to improve segmentation performance in our limited datasets with masked eyeball and optic nerve. We chose LiTS as the source dataset for the transfer learning of eyeball and optic nerve segmentation because it has proven effective in domain adaptation for orbital CT datasets with few samples [5].
These clarifications have been detailed in lines 480–484 of the manuscript.

Round 3
Reviewer 1 Report
Comments and Suggestions for Authors
The authors well addressed my concerns.